# Stratified medicine in European Medicines Agency licensing: a systematic review of predictive biomarkers

Kinga Malottki,[1] Mousumi Biswas,[2] Jonathan J Deeks,[1,3] Richard D Riley,[1,3] Charles Craddock,[4] Philip Johnson,[5] Lucinda Billingham[1,6]

For numbered affiliations see end of article

Correspondence to
Ms Kinga Malottki;
k.malottki@bham.ac.uk

## ABSTRACT

**Objectives:** Stratified medicine is often heralded as the future of clinical practice. Key part of stratified medicine is the use of predictive biomarkers, which identify patient subgroups most likely to benefit (or least likely to experience harm) from an intervention. We investigated how many and what predictive biomarkers are currently included in European Medicines Agency (EMA) licensing.

**Setting:** EMA licensing.

**Participants:** Indications and contraindications of all drugs considered by the EMA and published in 883 European Public Assessment Reports and Pending Decisions.

**Primary and secondary outcome measures:** Data were collected on: the type of the biomarker, whether it selected a subgroup of patients based on efficacy or toxicity, therapeutic area, marketing status, date of licensing decision, date of inclusion of the biomarker in the indication or contraindication and on orphan designation.

**Results:** 49 biomarker–indication–drug (B-I-D) combinations were identified over 16 years, which included 37 biomarkers and 41 different drugs. All identified biomarkers were molecular. Six drugs (relating to 10 B-I-D combinations) had an orphan designation at the time of licensing. The identified B-I-D combinations were mainly used in cancer and HIV treatment, and also in hepatitis C and three other indications (cystic fibrosis, hyperlipoproteinaemia type I and methemoglobinaemia). In 45 B-I-D combinations, biomarkers were used as predictive of drug efficacy and in four of drug toxicity. It appeared that there was an increase in the number of B-I-D combinations introduced each year; however, the numbers were too small to identify any trends.

**Conclusions:** Given the large body of literature documenting research into potential predictive biomarkers and extensive investment into stratified medicine, we identified relatively few predictive biomarkers included in licensing. These were also limited to a small number of clinical areas. This might suggest a need for improvement in methods of translation from laboratory findings to clinical practice.

## Strengths and limitations of this study

- Our research, to our knowledge, provides the first indication of the number and nature of predictive biomarkers included in licensing in Europe.
- We used systematic review methodology.
- It is likely that the 49 identified biomarker–indication–drug combinations do not represent a complete list of predictive biomarkers used in practice, as some could have been considered by national regulatory agencies, particularly for drugs considered before European Medicines Agency was established in 1995.

## INTRODUCTION

Drugs are rarely effective in all patients and may be associated with serious adverse events.[1] The challenge of stratified medicine is to identify predictive biomarkers that identify patient subgroups (or strata) with a differential therapeutic response to a linked intervention, allowing more appropriate and effective use of interventions to maximise patient benefit and minimise the occurrence of serious adverse events.[2][3] Predictive biomarkers are defined particularly to a treatment for a condition, where biomarker values are associated with differential efficacy or toxicity of that treatment.[4–7] The use of predictive biomarkers promises a more appropriate choice of treatment; it can also help to rationalise funding decisions, avoiding costs of futile treatment and of adverse events. However, the additional cost of measuring the marker has to be taken into account. Examples of predictive biomarkers include tamoxifen use in breast cancer, which is prescribed to women who are oestrogen receptor positive,[8] and trastuzumab, which is prescribed to those with HER2 overexpression in their tumour.[9]

There is a large body of literature documenting research into potential predictive biomarkers,[10][11] and millions of pounds have

been invested into stratified medicine, in industry and through programmes from funding bodies such as the Medical Research Council[12] and Cancer Research UK.[13] We aimed to investigate whether this interest in developing stratified medicines has led to production of biomarker-treatment combinations ready for use in clinical practice. To explore this question, we have undertaken a systematic review of predictive biomarkers reported in licensing decisions of the European Medicines Agency (EMA).

In our review, we aimed to find out how many of the indications and contraindications considered by the EMA define a patient population using a predictive biomarker. We were also interested in the disease areas where predictive biomarkers have been used and any trend over time. It has been hypothesised that stratified medicine has not been implemented in practice as much as expected. This paper provides evidence of the impact of stratified medicine research to date and if less than expected, then this will highlight the need to review the underlying reasons and address the problems.

## METHODS

We defined a biomarker–indication–drug (B-I-D) combination as the unit of our analysis, relating to the use of a predictive biomarker with a particular drug for a particular condition or disease.[4–7] For toxicity biomarkers where biomarkers of drug toxicity may be used in more than one disease area we grouped these into one B-I-D combination.

All drugs listed on the EMA website in either European Public Assessment Reports or Pending Decisions[14 15] (accessed on 17 January 2013) were evaluated, together with their indications and contraindications.

Our inclusion criteria were that the biomarker had to
1. Be used in the indication and/or contraindication of a drug;
2. Be associated with a particular treatment;
3. Identify a subgroup of patients with a particular disease eligible for treatment with the drug.

We excluded biomarkers
1. Associated with a non-therapeutic substance (eg, vaccines).
2. Not used as predictive, including:
    A. Used for diagnosis, screening or forming part of the disease definition (already established for defining a disease) or established disease subtype;
    B. Prognostic only (associated with outcome regardless of treatment and not predictive of treatment response[16]).
3. Associated with another treatment (eg, the biomarker was not associated with the differential efficacy or toxicity of the drug of interest, but another drug given in combination with the drug of interest).

We have reviewed EMA licensing, as in Europe, a centralised drug evaluation by the EMA is required for drugs used in the treatment of a number of conditions, drugs obtained from biotechnology processes and all drugs used for rare conditions (orphan medicines). Companies can also apply for a centralised marketing authorisation of other drugs.[17] Although the EMA does not license biomarkers, it evaluates drugs in groups defined by predictive biomarkers (eg, trastuzumab is licensed for use in HER2 overexpressing patients with breast cancer).[18] Our approach is likely to give a broad overview of the impact of predictive biomarkers on treatment selection since 1995 (when EMA was established[19]).

We created a database of all drugs in the EMA database including the drug name, licensing status, indication and contraindication. In the first stage of screening, all database entries were screened by two independent reviewers (MB and KM) to identify those potentially including a predictive biomarker in the indication or contraindication. If an entry was identified by at least one of the reviewers as potentially relevant, it was included in the second stage of screening.

In the second stage of screening, a list of potential B-I-D combinations was created based on the entries identified in the first stage. The list of potential B-I-D combinations was assessed by two independent reviewers (MB and KM) using full inclusion/exclusion criteria, based on the information in the Summary of Product Characteristics (which sets out the position of the drug obtained in the assessment process and summarises its properties and clinical use together with the clinical trial evidence that was considered by the EMA),[20] the Scientific Discussion (which discusses the properties and clinical evidence in more detail) and additional information from targeted internet searches and expert advice if necessary. Any disagreements were resolved by discussion.

For the included B-I-D combinations, data were collected on: the type of the biomarker used as predictive, whether it selected a subgroup of patients based on efficacy or toxicity, therapeutic area, marketing status, date of licensing decision, date of inclusion of the biomarker in the indication or contraindication and on orphan designation (granted to drugs intended for the treatment of a life-threatening or chronically debilitating condition which is either affecting no more than 5 in 10 000 people in the European Union or when the revenue is unlikely to cover the investment in drug development[21]). To provide a context for our review, we have also collected data on the total number of drugs licensed each year with and without an orphan designation.

## RESULTS

Across the 18-year period (1995–2012), we identified 49 B-I-D combinations, including 37 biomarkers and 41 different drugs. The details of the review process are

presented in figure 1. Most of the drugs were authorised, the exceptions being

▶ Gemtuzumab ozogamicin (refused);
▶ Zeldoronic acid (pending);
▶ Imatinib in the indication for aggressive systemic mastocytosis (withdrawn);
▶ Amprenavir (withdrawn);
▶ Nelfinavir (withdrawn).

The number of new B-I-D combinations considered by the EMA each year has increased overall from 0 to 1/year in the late 1990s, to a maximum of 7 in each of 2011 and 2012 as shown in figure 2. This was, however, not a steady increase, as the number of B-I-D combinations considered by the EMA showed a fluctuation between 2000 and 2006, a decrease between 2006 and 2010, followed by an increase in the number in 2011 and 2012. A predictive biomarker was included in the

indication or contraindication at the time when the drug was first licensed for 35 drugs (for one (capecitabine), the date of inclusion of the biomarker was unclear from the documentation; for the remaining drugs, the time from the initial licensing decision to the inclusion of a predictive biomarker ranged from 1 to 10 years). The proportion of first licensing decisions of all new drugs that included a predictive biomarker increased over time and was close to 10% in 2003, 2004, 2005, 2011 and 2012 (figure 3).

Six drugs associated with a predictive biomarker had an orphan designation at the time of licensing; however, for two drugs, it was removed at the end of exclusivity period (details reported in table 1). One of the six drugs (imatinib) was associated with five different predictive biomarkers in five different indications (figure 4).

The identified predictive biomarkers were all molecular. Thirty-three biomarkers were used to predict treatment efficacy (details reported in table 1) and only four to predict toxicity (table 2).

Most of the biomarkers were included in indications and contraindications of cancer treatments (26 B-I-D combinations) and viral diseases, mainly HIV (17 B-I-D combinations). The remaining biomarkers were used to stratify metabolic and blood disorders (cystic fibrosis, hyperlipoproteinaemia type I and methemoglobinaemia) and appeared in the past 2 years (figure 2).

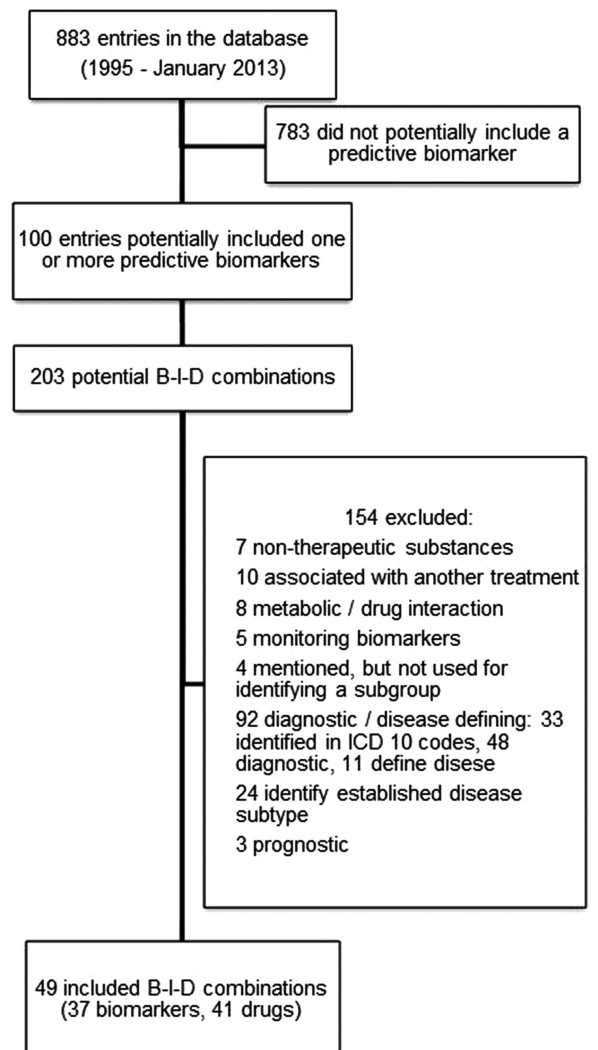

**Figure 1** Flow diagram for the systematic review of predictive biomarkers in the European Medicines Agency licensing. B-I-D, biomarker–indication–drug; ICD, International Classification of Diseases.

## DISCUSSION

Stratified medicine is promoted as key to the future of medicine, and is currently one of the most active areas of clinical research. To our knowledge, this review provides the first indication of the number and nature of predictive biomarkers included in licensing in Europe based on the drug indications and contraindications on the EMA website. Forty-nine B-I-D combinations were identified. All identified biomarkers were molecular. The identified B-I-D combinations were mainly used in cancer and HIV treatment, with only five used in other disease areas.

It is likely that the 49 identified B-I-D combinations from the EMA database do not represent a complete list of the predictive biomarkers used in practice as some predictive biomarkers could have been considered by national regulatory agencies, particularly for drugs considered before EMA was established in 1995. Also, EMA licensing is not compulsory for some disease areas, such as mental health. However, a number of drugs with indications in depression of schizophrenia have been considered by the EMA. Therefore, we believe that although our approach might not provide a complete list of all predictive biomarkers used in Europe, relatively few are likely to have been omitted, particularly from recent years.[19] The fact that some of the identified B-I-D combinations included biomarkers introduced to an indication of an already licensed drug suggests that at least to some extent we have captured stratification occurring after the

**Figure 2** Number of new biomarker–indication–drug (B-I-D) combinations considered each year by disease area (includes biomarkers added after the drug was initially licensed).

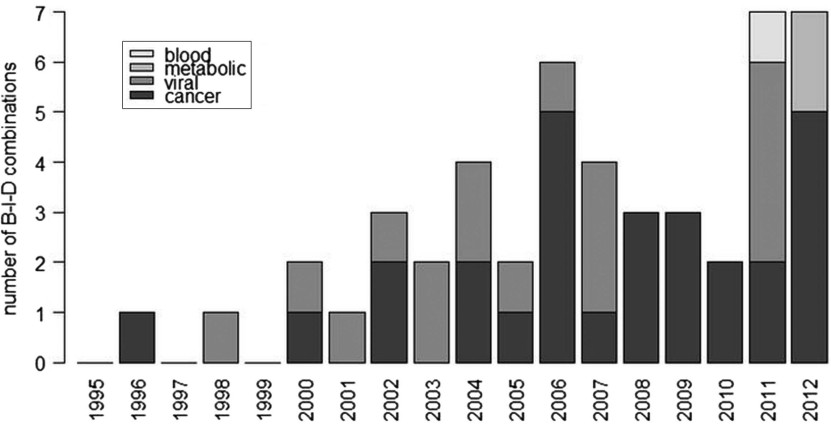

initial licensing of a drug. However, the actual extent to which this takes place in clinical practice is difficult to evaluate.

Several types of biomarkers were excluded. We did not include biomarkers used for dose adjustments as they do not directly predict efficacy or toxicity (although inappropriate dose adjustment could limit the treatment efficacy or cause adverse events).[22] We also investigated only biomarkers associated with drug treatments. Other biomarkers may be used in practice with non-drug treatments (eg, radiotherapy).

The definition of a predictive biomarker can be difficult to apply, as over time predictive biomarkers may become part of a redefinition of the disease or subtype of disease[23] and be classed as diagnostic tests. In our evaluation, we excluded diagnostic biomarkers (eg, factor IX deficiency, or genetic testing for familial lipoprotein lipase deficiency) and biomarkers used to identify an established subtype of a disease (mainly ST segment elevation and non-ST segment elevation myocardial infarction).

The spectrum of diseases where predictive biomarkers have been successfully developed is relatively narrow. This suggests a possible need for more research in other clinical areas. Also, the vast majority of the B-I-D combinations were associated with treatment efficacy and only four with toxicity. As adverse events associated with some treatments could be potentially serious and the possibility to screen out patients at high risk prior to the start of the treatment would be beneficial. A proportion of the drugs with an associated predictive biomarker identified in our review had an orphan designation. This seems surprising, as convincing evidence to support the use of a drug in a subgroup of patients with a rare condition might be difficult to obtain, due to the small numbers of patients available to test the hypotheses.

It is difficult to provide accurate estimates of the extent of research into potential predictive biomarkers; however, it has been suggested in 2011 that the number of publications on different biomarkers (not only predictive) was around 15 000.[10] Another paper published in 2009, which reviewed genetic markers evaluated as potential predictors

**Figure 3** New drugs authorised each year with and without a predictive biomarker in the indication or contraindication (excludes biomarkers added after the drug was initially licensed).

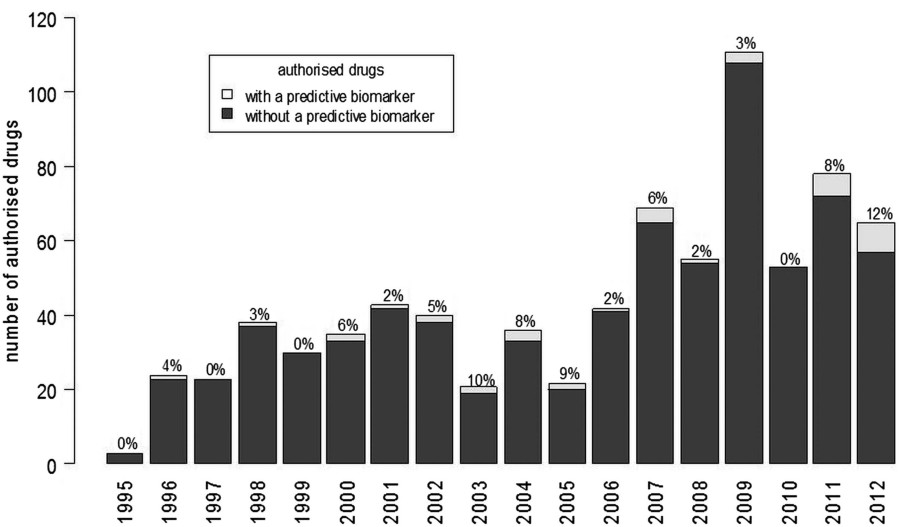

**Table 1** Biomarkers predictive of efficacy identified in the review of European Medicines Agency licensing

| Biomarker | Indication | Drug |
| --- | --- | --- |
| ALK gene rearrangement | Carcinoma, non-small-cell lung | Crizotinib (Xalkori) |
| BRAF V600 mutation | Melanoma | Vemurafenib (Zelboraf) |
| CCR5 tropism | HIV infections | Maraviroc (Celsentri) |
| CD-33 expression* | Leukaemia, myeloid, acute | Gemtuzumab ozogamicin (Mylotarg)† |
| EGFR expression | Colorectal neoplasms | Cetuximab (Erbitux) |
| EGFR expression | Carcinoma, non-small-cell lung | Erlotinib (Tarceva) |
| EGFR mutation | Carcinoma, non-small-cell lung | Erlotinib (Tarceva) |
| EGFR mutation | Carcinoma, non-small-cell lung | Gefitinib (Iressa) |
| EpCAM expression | Cancer ascites | Catumaxomab (Removab) |
| FIP1L1-PDGFR rearrangement | Hypereosinophilic syndrome | Imatinib (Glivec)‡ |
| G551D mutation in the CFTR gene | Cystic fibrosis | Ivacaftor (Kalydeco)† |
| Genotype 1 HCV | Hepatitis C, chronic | Boceprevir (Victrelis) |
| Genotype 1 HCV | Hepatitis C | Telaprevir (Incivo) |
| HER2 expression | Breast neoplasms | Lapatinib (Tyverb) |
| HER2 expression | Breast neoplasms | Trastuzumab (Herceptin) |
| HER2 expression | Stomach neoplasms | Trastuzumab (Herceptin) |
| HER2 expression | Breast neoplasms | Everolimus (Afinitor) |
| HER2 expression§ | Breast neoplasms | Pertuzumab (Perjeta) |
| Hormone dependency | Prostatic neoplasms | Degarelix (Firmagon) |
| Hormone receptor expression§ | Breast neoplasms | Zoledronic acid (Zometa) |
| Hormone receptor expression | Breast neoplasms | Everolimus (Afinitor) |
| Kit (CD 117) expression | Gastrointestinal stromal tumours | Imatinib (Glivec)‡ |
| Kit (D816V) mutation¶ | Aggressive systemic mastocytosis | Imatinib (Glivec)‡ |
| KRAS mutation | Colorectal neoplasms | Cetuximab (Erbitux) |
| KRAS mutation | Colorectal neoplasms | Panitumumab (Vectibix) |
| LPL protein detectable | Hyperlipoproteinaemia type I | Alipogene tiparvovec (Glybera)† |
| Oestrogen receptor expression | Breast neoplasms | Fulvestrant (Faslodex) |
| Oestrogen receptor expression | Breast neoplasms | Toremifene (Fareston) |
| PDGFR gene rearrangements | Myelodysplastic-myeloproliferative diseases | Imatinib (Glivec)‡ |
| Philadelphia chromosome | Precursor cell lymphoblastic leukaemia-lymphoma | Dasatinib (Sprycel)† |
| Philadelphia chromosome | Precursor Cell lymphoblastic leukaemia-lymphoma | Imatinib (Glivec)‡ |
| t(15; 17) translocation | Leukaemia, promyelocytic, acute | Arsenic trioxide (Trisenox)‡ |
| Viral resistance mutations¶ | HIV infections | Amprenavir (Agenerase) |
| Viral resistance mutations | HIV infections | Atazanavir sulfate (Reyataz) |
| Viral resistance mutations | HIV infections | Darunavir (Prezista) |
| Viral resistance mutations | HIV infections | Efavirenz/emtricitabine/tenofovir disoproxil (Atripla) |
| Viral resistance mutations | HIV infections | Emtricitabine (Emtriva) |
| Viral resistance mutations | HIV infections | Emtricitabine/rilpivirine/tenofovir disoproxil (Eviplera) |
| Viral resistance mutations | HIV infections | Enfuvirtide (Fuzeon) |
| Viral resistance mutations | HIV infections | Fosamprenavir calcium (Telzir) |
| Viral resistance mutations | HIV infections | Lopinavir/ritonavir (Kaletra) |
| Viral resistance mutations¶ | HIV infections | Nelfinavir (Viracept) |
| Viral resistance mutations | HIV infections | Rilpivirine hydrochloride (Edurant) |
| Viral resistance mutations | HIV infections | Tenofovir disoproxil fumarate (Viread) |
| Viral resistance mutations | HIV infections | Tipranavir (Aptivus) |

*Refused.
†Drug designated an orphan medicine.
‡Orphan designation has been removed at the end of exclusivity period.
§Pending.
¶Withdrawn.

**Figure 4** New orphan drugs authorised each year with and without a predictive biomarker in the indication or contraindication (excludes biomarkers added after the drug was initially licensed).

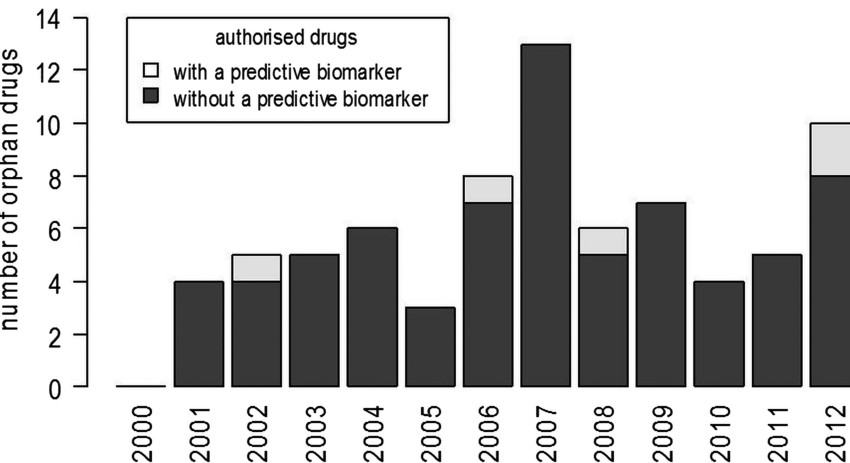

of response to treatment, found that 541 different genes were investigated as potential predictive biomarkers in 1668 papers.[11] It can be reasonably expected that this number largely increased since these papers were published. Our review shows that few predictive biomarkers have been included in licensing relative to this large body of literature documenting numerous potential predictive biomarkers. Therefore, in spite of the substantial investment in research, the promise of stratified medicine is not yet being realised to a large extent. The reasons for this might include poor translation of findings of laboratory studies into clinical context, or the failure to identify effective predictive biomarkers and treatments. Even though it is becoming easier and cheaper to gather huge sets of genomic data, its interpretation is challenging, which can potentially hinder translational research. Recognising this, initiatives have been undertaken in the USA (National Institutes of Health and the Food and Drug Administration) and the UK (Medical Research Council) to promote the translation of basic research into clinical practice.[12] In addition, the availability of datasets such as the Cancer Cell Line Encyclopaedia and a similar UK initiative might contribute to the faster progress of stratified medicine.[24 25] The relatively small number of predictive biomarkers identified in licensing might also indicate the need for more sound methodological standards for biomarker discovery and development.[26]

**Author affiliations**
[1]MRC Midland Hub for Trials Methodology Research, University of Birmingham, Birmingham, UK
[2]The Discovery Research Programme, School of Social and Community Medicine, University of Bristol, Bristol, UK
[3]Department of Public Health, Epidemiology and Biostatistics, School of Health and Population Sciences, University of Birmingham, Birmingham, UK
[4]Centre for Clinical Haematology, Queen Elizabeth Hospital, Birmingham, UK
[5]University of Liverpool & Clatterbridge Cancer Centre NHS Foundation Trust, Liverpool, UK
[6]Cancer Research UK Clinical Trials Unit, University of Birmingham, Birmingham, UK

**Contributors** KM, JJD, RDR and LB designed the review. KM and MB carried out the review. Where needed CC and PJ provided clinical advice. All authors contributed to the interpretation of the results, commented on drafts and accepted the final version of this paper.

**Funding** This work was funded by the MRC Midlands Hub for Trials Methodology Research at the University of Birmingham (Medical Research Council Grant ID G0800808).

**Competing interests** None.

**Provenance and peer review** Not commissioned; externally peer reviewed.

**Data sharing statement** Full dataset available from the corresponding author on request.

**Table 2** Biomarkers predictive of toxicity identified in the review of European Medicines Agency licensing

| Biomarker | Indication | Drug |
|---|---|---|
| DPD deficiency | Colorectal neoplasms<br>Colonic neoplasms<br>Stomach neoplasms<br>Breast neoplasms | Capecitabine (Xeloda and generic drugs: capecitabine accord; capecitabine krka; capecitabine medac and Capecitabine teva) |
| DPD deficiency | Stomach neoplasms | Tegafur/gimeracil/oteracil (Teysuno) |
| HLA-B*5701 allele | HIV infections | Abacavir (Kivexa; Trizivir and Ziagen)* |
| NADPH reductase deficiency | Methemoglobinaemia | Methylthioninium chloride (Methylthioninium chloride Proveblue) |

*HLA-B*5701 allele is predictive of hypersensitivity to abacavir, which is present in three drugs: Kivexa (abacavir/lamivudine); Trizivir (abacavir/lamivudine/zidovudine) and Ziagen (abacavir).

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
