## [Reviewer comments · BMJ Open]

Some articles will have been accepted based in part or entirely on reviews undertaken for other BMJ Group journals. These will be reproduced where possible.

ARTICLE DETAILS

TITLE (PROVISIONAL)	Stratified medicine in European Medicines Agency licensing: a systematic review of predictive biomarkers
AUTHORS	Malottki, Kinga; Biswas, Mousumi; Deeks, Jon; Riley, Richard; Craddock, Charles; Johnson, Philip; Billingham, Lucinda

VERSION 1 - REVIEW

REVIEWER	Mahesh Parmar and Louise Brown Medical Research Council Clinical Trials Unit at UCL
REVIEW RETURNED	12-Nov-2013

GENERAL COMMENTS	Overall, this is a well written paper that aims to assess and document the extent to which biomarkers that might predict response to specific treatments have been registered for licensing within the EMA since it was established in 1995. We are not sure why they have selected the EMA database alone as their source. This is an important question in general but the authors have restricted their sampling to the Biomarker-Indication-Drug (B-I-D) combinations reported in the European licensing context. As a result, we are not sure they can make any broad conclusions about the current extent of predictive biomarker use, particularly the conclusions they draw in the closing paragraph of the discussion as the use of biomarkers in treatment may not always be of interest to companies registering for licensing purposes. They should take care to always discuss their work in relation to the European regulatory framework. For example, on page 5, line 32-33 they make this statement: “Our approach is likely to give a broad overview of the impact of predictive biomarkers on treatment selection since 1995 (when EMA was established). “ There is an ever-widening field of biomarker use documented in peer reviewed journals, much of which will never make it to the EMA database. The authors have eluded to this by citing reference 10 and they do refer to it as a limitation in their discussion but they do not completely explain the rationale for their study design. The authors need to discuss the full implications of taking the sampling frame that they have. On the whole, their methods are sound (if their hypothesis was only to review what was entered into the EMA database). Their approach is methodical and appears to be conducted according to PRISMA requirements but the page numbers they provide on the PRISMA checklist do not tally up with the manuscript pages. Other comments are:
--

	1) The methods could be more detailed. We are not entirely clear of their search or filtering criteria. They say they used the Summary of Product Characteristics, the Scientific Discussion and any other information from targeted internet searches. Did they really read all this documentation for all 883 entries in the EMA database? Was there no simple filtering by title or summary alone first? They managed to eliminate 783 entries as “Not potentially including a predictive biomarker”. What were the main reasons for exclusion (in terms of their exclusion criteria listed on page 5)? These would normally be documented as an appendix in a systematic review. 2) In figures 2 and 3, is there a discrepancy in the year 2010? Figure 2 suggests 2 biomarkers were considered but Figure 3 suggests 0% for that year. Perhaps we have misunderstood the graphs? 3) For completeness in Figures 3 and 4 the authors should provide a legend describing the column colours. 4) On page 6, lines 35-37, the authors list the years where the proportion approached 10%. This was also the case in 2005.
--	--

REVIEWER	Amy Mulick Cassidy London School of Hygiene and Tropical Medicine, Medical Statistics
REVIEW RETURNED	15-Nov-2013

GENERAL COMMENTS	This is a small study using publicly available data from the EMA to identify the number and types of drugs that are approved on biomarker status are currently licenced in the EU. The abstract defines the study this way, but the authors have stated a stronger hypothesis in the introduction which is that they will investigate whether increased investment into stratified medicine has led to the approval of more biomarker-treatment combinations in clinical practice. However, no statistical analysis was undertaken so it is not possible for this study to offer evidence of a causal link. If this is what the authors wish to show, I would suggest splitting the data into pre- and post-investment time spans and using a poisson regression with number of new biomarker drugs as the outcome to test this, perhaps with a lag to account for development time. This would strengthen the paper. Figures 2 and 3 seem to contradict each other; some percentages in Fig 3 do not match with the numbers represented in Fig 2 (e.g. 2006 and 2010). Perhaps this is related to issues below. The authors state that the proportion of first licensing decisions of new drugs with a predictive biomarker increased over time from 1995. This is slightly misleading; the proportion roughly increased in the 8 years from 1995-2003, but then roughly decreased in the following 7 years, reaching 1995 levels by 2010. It is not clear whether the higher proportions in 2011 and 2012 are evidence of a trend to growth again or whether they are anomalous. However, I am not sure this is the best way to capture growth in stratified medicine. First licensing statistics ignore existing drugs that are approved for further uses later in life, and indeed it appears that only 30-35 of the 49 BID combinations in Fig 2 are represented in Fig 3. It would be better to report this as a proportion of all licencing decisions. Clarifying when investment was increased and providing any available information on the expected lag between investment and drug approval would also put these figures in a more useful
--

	perspective. There was no mention of this in the discussion, but it can take a very long time to assess and validate potential biomarkers; especially to obtain sufficient numbers of patient biosamples and to develop and validate appropriate and reliable assays. Finally, the last paragraph claims to have shown that ‘few predictive biomarkers have been included in licensing relative to the large body of literature documenting numerous potential predictive biomarkers.’ The extent of the potential biomarkers was never illustrated – further information from refs 10 and 11 would be required to make this claim. Conclusion: this is a nice article and it will be a useful addition to the evidence base on translating research in this area into clinical practice. The preceding comments were minor and offered in a constructive spirit.
--	--

VERSION 1 – AUTHOR RESPONSE

Reviewers 1 have asked about the rationale and consequences of only searching the EMA website. “We are not sure why they have selected the EMA database alone as their source. (...) The authors need to discuss the full implications of taking the sampling frame that they have”

We have amended the discussion accordingly to provide more information. Please see page 7, paragraph 2 where we discuss the reasons for and limitations of the approach we have taken.

They have also asked for more clarity regarding the filtering criteria.

“Was there no simple filtering by title or summary alone first?”

We have therefore extended the methods section accordingly to provide more detail (please see page 5). We have initially made decisions on the possible presence of the biomarker based on the text of the indications and contraindications.

The reviewers also commented on the fact that the page numbers in the PRISMA checklist did not tally up with the manuscript pages. This was due to the fact that pages were re-numbered in the document generated by BMJ Open system.

Reviewer 2 suggested that we have put too much emphasis on funding issues and our study does not assess the impact of funding decisions.

“the authors have stated a stronger hypothesis in the introduction which is that they will investigate whether increased investment into stratified medicine has led to the approval of more biomarker-treatment combinations in clinical practice. However, no statistical analysis was undertaken so it is not possible for this study to offer evidence of a causal link.”

Our aim was never to investigate the influence of funding decisions. The information on funding provided in the introduction was simply to provide a general overview of the activity in the stratified medicine field. We have modified the introduction to avoid any further misunderstandings by changing a statement which may have sounded too strong and is now:

“We aimed to investigate if this interest in developing stratified medicines has led to production of biomarker-treatment combinations ready for use in clinical practice.”

Reviewer 2 also raised the issue of post-marketing changes in drug indications.

“licensing statistics ignore existing drugs that are approved for further uses later in life”

This has now been addressed in the discussion (please see page 7).

Reviewer 2 also asked for more detail on the literature on potential biomarkers to be included:

“Finally, the last paragraph claims to have shown that ‘few predictive biomarkers have been included in licensing relative to the large body of literature documenting numerous potential predictive biomarkers.’ The extent of the potential biomarkers was never illustrated – further information from refs 10 and 11 would be required to make this claim.”

This has now been added to the discussion (page 8).

Reviewers 1 and 2 made some comments regarding consistency and labelling of figures.

Reviewers 1:

“In figures 2 and 3, is there a discrepancy in the year 2010? Figure 2 suggests 2 biomarkers were considered but Figure 3 suggests 0% for that year. Perhaps we have misunderstood the graphs?”

“For completeness in Figures 3 and 4 the authors should provide a legend describing the column colours.”

Reviewer 2:

“Figures 2 and 3 seem to contradict each other; some percentages in Fig 3 do not match with the numbers represented in Fig 2 (e.g. 2006 and 2010).”

Our figures have now been changed to provide more detail and consistency as requested. We have added legends to Figure 3 and 4 and added an explanatory text to the caption of Figure 2.

Reviewers 1 and 2 also commented on the incompleteness of the description of the trend over time.

Reviewers 1:

“On page 6, lines 35-37, the authors list the years where the proportion approached 10%. This was also the case in 2005.”

Reviewer 2:

“The authors state that the proportion of first licensing decisions of new drugs with a predictive biomarker increased over time from 1995. This is slightly misleading; the proportion roughly increased in the 8 years from 1995-2003, but then roughly decreased in the following 7 years, reaching 1995 levels by 2010. It is not clear whether the higher proportions in 2011 and 2012 are evidence of a trend to growth again or whether they are anomalous.”

This has been amended accordingly on page 6, where we have provided a more detailed discussion of the trend.